# Analysis of the Degradation of Pearlitic Steel Mechanical Properties Depending on the Stability of the Structural Phases

**DOI:** 10.3390/ma16020518

**Published:** 2023-01-05

**Authors:** Radim Šmak, Jiří Votava, Jaroslav Lozrt, Vojtěch Kumbár, Tomáš Binar, Adam Polcar

**Affiliations:** 1Department of Technology and Automobile Transport, Faculty of AgriSciences, Zemědělská 1, Mendel University in Brno, 613 00 Brno, Czech Republic; 2Department of Electrotechnology, Faculty of Electrical Engineering and Communication, Brno University of Technology, Technická 3058/10, 616 00 Brno, Czech Republic

**Keywords:** heat treatment, pearlite steel, carbon steel, carbide, hardness, acoustic emission

## Abstract

The paper is focused on analysing the pearlitic steel phase transformations and their influence on the mechanical properties. The intention is to perform a detailed analysis of the heat treatment process using the exact heating temperature and chemical composition to achieve the optimal mechanical properties of the tool. The key area is monitoring and regulating the heat treatment. This technology is constantly undergoing an optimisation process and is an effort to introduce new trends in monitoring phase transformations and processes. The use of non-destructive methods is an adequate tool. The principle is to determine the exact structural phase at a given moment, which can be very difficult when a complex shaped part is heat treated. Which precludes the use of some other methods of phase transformation analysis. Specifically, the determination of the exact moment of finish of the austenitisation process is eminent. The monitoring of these processes will be ensured by both a non-contact pyrometer and also by the AE method with an adequate sensor and waveguide. The resulting structural phases formed after the heat treatment will be evaluated by electron microscopy, followed by the analysis of the mechanical properties of selected steels.

## 1. Introduction

A properly performed heat treatment of steel can achieve adjustments to the qualities of toughness, hardness, abrasion resistance, etc. [1]. The optimisation of the process and the correct recrystallisation temperature are always a problem, caused by the heat-treated component shape [2]. The steel sensitivity to heat treatment primarily depends on the carbon content. In addition to the main alloying elements, such as carbon, silicon, and manganese, ref. [3] mentions chromium, vanadium, and especially nitrogen as elements that significantly affect the mechanical properties of steel, especially at low temperatures. As another element that significantly influences the mechanical properties, ref. [3] mentions molybdenum, which ensures an increase in the toughness. Carbide-forming elements also play a significant role in the internal arrangement of the microstructure. Sulfur content significantly affects the structure and properties of steel. Sulfur enters the steel during pig iron production or as an artificially added alloying element [4]. Based on the exact chemical composition and the change in the physical properties during the heating time delay, the moment of the crystal lattice transformation can be optimised and, thus, the maximum potential of the individual chemical elements in the material can be used. The team of authors [5] clearly confirm that the heating time significantly affects the microstructure and mechanical properties of carbon spring steel. The authors also report a significant influence of the heating method of the part. When electrical induction heating is used, the crystal lattices are affected and the resistance to dislocation motion is reduced. Using the change in the crystal lattice’s physical properties during the heat treatment process is one of the ways to eliminate its defects [6]. Monitoring the process parameters is a very important aspect of heat treatment [7]. The processes of iron alloy phase transformations can be effectively monitored using acoustic emission methods [3]. Paper [8] mentions an acoustic emission analysis, among other methods, as a suitable means of detecting the ongoing changes within the material, e.g., phase transformations. Cracking and similar processes in the material must be analysed separately, due to the difference in amplitude and frequency compared to phase transformations. Due to the high acoustic emission (AE) frequency, the interfering low frequency AE signals can be effectively filtered out. This paper focuses mainly on the analysis of the high frequency signal produced by phase transformations in steels. In this paper, the hit-based collection method was used. The hit-based method is suitable for capturing signals arising from cracks in the material. During the data collection, it is necessary to set the threshold (amplitude) value to limit the majority of the background noises [9]. Amplitude thresholds are suitable for a various frequency analysis. Therefore, a wide range of frequencies can be recorded. The data are then processed using a PSD analysis. The process of martensitic transformation can also be monitored using the Barkhausen noise emission. Barkhausen noise is caused by internal stress fields and depends on the steel microstructure [10]. The author [11] states that the frequency of 20–900 kHz was used for the AE probe and the converter sampling frequency of 1.2 MHz was used for the analysis of phase transformations below Ms. On the basis of previous experience and according to the research processing issue, it is possible to predict the development and research focused on austenitisation criteria.

The primary research goal is to characterise the areas of the martensitic structural decay and austenitisation during heat treatment for the selected steels. Furthermore, the goal is to define this area without any unnecessary time delay, which is accompanied by the occurrence of material defects, but by also changing the entire crystalline planes, which manifests itself in grain growth. A number of factors affect the grain growth process: temperature high above A3, inadequate heating time, high dislocation density, rolling texture and others [12].

## 2. Materials and Methods

The phase transformation analysis was performed using acoustic emission methods at the recrystallisation temperatures of the steel (830 °C for the C45 steel and 800 °C for the C105W1 and 80CrV5 steels). The following steels were selected for the research purposes: C45, C105W1, and 80CrV5; all steels were commercially produced. The C45 steel is close to a eutectoid composition with a high pearlitic content in its natural state (see Figure 1). The dispersity of perlite is high according to the carbon content of this steel. In addition to perlite, C45 steel contains ferrite in the microstructure. C105W1 contains a primarily perlitic structure with a higher proportion of iron carbide. In addition to the perlitic structure, 80CrV5 contains chromium-based superdeutectoid carbides of varying chemical compositions. The C105W1steel was selected as a carbon tool steel with a carbon content of 1%, and the 80CrV5 steel was chosen a as carbon tool steel alloyed with chromium. The chemical composition of the used steels is shown in Table 1. The table contains both the values given by the material standard and the spectroscopically measured values. The sulphur and phosphorus content has not been analysed.

The selected steels are widely used in the industry as a material for the production of a wide range of engineering products, such as forming and woodworking tools. Significant volumes of similar products are heat-treated in these plants. For this reason, great emphasis is placed on the most efficient heat treatment process in the shortest possible time, whether due to high production efficiency or ecology.

### 2.1. Testing System

According to the needs of the experiment, test specimens having a diameter of 7 mm and a length of 20 mm were created from the selected steels. All samples were made from hot rolled flat bars. The flat bars were subjected to an annealing heat treatment to reduce stresses before specimen preparation, with temperatures according to the material standard. The sample dimensions were chosen for further analysis. The material was analysed mainly on a microscopic level in the form of metallographic preparations.

The basic heat treatment was performed on a premade test set (see Figure 2) consisting of a steel sample placed on a waveguide made of austenitic stainless steel, which does not have its own phase transformation in the monitored temperature range from room temperature to 800 °C or 830 °C. The heat treatment was performed in an electric laboratory furnace MP 05–1.1. This device offers the possibility of placing a sample connected to the waveguide, where the AE probe is placed outside of the heated area of the furnace (see Figure 2). The connection with the waveguide was ensured by a fusion weld using the tungsten inert gas (TIG) method. For the experiment, a Dakel XEDO 080 AE (4 MHz sampling, Hořovice, Czech Republic) unit was used. The assembly was connected to the AE probe via a 35 dB amplifier. An AE probe IDK 09 (20–1 MHz) with a corundum surface layer was used for the experiment. In the acoustic emission spectrograms, the amplitude is given in mV. Due to the nature of the analysed processes, the hit mode EA measurement technique was used. A target with an acoustic emission signal probe was also welded at the end of the waveguide. The entire test set-up was introduced into the furnace using a feeding device, which guaranteed the same start time of the measurement. The temperature of the waveguide and AE probe was maintained at a room temperature of 22 °C with a maximum deviation of 5 °C. The temperature measurement of the AE probe was provided by a non-contact thermometer. In order to avoid unwanted thermal effects caused by welding the waveguide to the test specimen, a sample area of 12 mm from the free end was defined. This distance was metallographically verified. The accurate measurement of the heating times was critically necessary for the further research. The sample—wave guide—AE probe test set was used in an experiment performed by [13], when an acoustic emission signal generated by the formation of a martensitic structure during hardening was monitored.

The basis of the AE analysis methodology is the detection of elastic waves created by specific processes in the material, such as phase transformations, crack formations and others [8]. The measurement of the acoustic emission signal was performed, in this case, by the passive method. The measuring set must not have any significant effect on the analysed sample. When this condition is met, the captured signals relate only to the acoustic emission events occurring in the monitored material [13].

During the experiment, the changes caused by the change in delay at the austenitisation temperature were monitored. Due to the increasing heating time, significant changes occur, which are evident in both the microstructure and mechanical properties of the steel. These are mainly grain coarsening and the degradation of alloying elements (burning) and undesirable decarburization of the surface layer. All of this results in a reduction in the mechanical properties of the used steels. These changes can be effectively verified by mechanical property tests (hardness tests, microhardness tests) and metallographic analysis.

### 2.2. Sample Heating

The heating of the samples was carried out in an electric resistance furnace with the possibility of controlling the temperature in 0.1 °C increments. The heating temperature was selected according to the A3 or ACM curves of the steel (C45–A3 780 °C, C105W1–ACM 750 °C, 80CrV5–ACM 745 °C). To verify the acoustic emission measurements, the time dependence of the sample’s core temperature was determined. The measurements were performed with a thermocouple with an adequate compensation line. Two sets of measurements were performed for heating the samples to 830 °C (the C45 steel) and 800 °C (the C105W1 and 80CrV5 steels). The temperatures were selected according to the material standard.

## 3. Results and Discussion

### 3.1. Heating Time

From the measurements, it can be concluded that the sample’s core reaches the austenitising temperature in 120–140 s. The heat transfer process is plotted graphically in Figure 3. As the heating rate increases, the temperature hysteresis between the surface and the component’s core will increase during the heating process. However, these temperatures will level off during the delay period. A risk factor is the development of internal stresses due to temperature differences in the core and surface layers. Based on the developed mathematical model, the exact values of the sample temperatures can be derived as a function of the heating time. After the time of 120/140 s is reached, the temperature in the furnace does not increase anymore, and the subsequent time delay takes place at a constant temperature.

### 3.2. Analysis of the Phase Transformations Using AE

Acoustic emission signals can be recorded and analysed during the phase transformations of carbon steel [13]. AE is a suitable tool for the design of the optimal heat treatment and microstructure shaping [11,14]. The physical basis of the acoustic emission event during phase transformations lies in the deformation processes and the structural transformations [15]. The basis of the research was the use of non-destructive acoustic emission methods to accurately define the moment of each phase transformation. The measurement output is a graphical representation of the high frequency acoustic signal produced by the phase transformations in the heat loaded steel. Before the heat treatment used for the experiment, all of the used steels were found to have a predominantly martensitic structure with carbides. Based on a previous analysis, the following methodology was chosen: the aim of the research was to heat load the specimens with a pre-formed martensitic structure. Thus, a more distinct and clearer acoustic emission response can be achieved. From this unbalanced structure, it is possible to trace the progress of other structures at a precise point in time. Primarily, it is necessary to know the beginning and especially the end of austenitisation. This time instant is a critical factor for controlling any further heat treatment. The correlation between Figure 3 is very important for this research, in which the measured values of the sample temperatures and the acoustic emission responses are shown. On the horizontal axis, time [s] is plotted for these graphical dependencies. Therefore, it is possible to relate the times of the acoustic emission events to a specific temperature and vice versa.

In contemporary publications, research on acoustic emission signals is based on the analysis of above-average signals and on the analysis of AE events [16]. Therefore, the start of the phase transformation was defined as a 20% steady-state increase in the acoustic emission event compared to the steady-state value.

The intensity of the acoustic emission event is strongly influenced by the carbon content of the steel. For all the steels analysed, two basic areas can be clearly distinguished. After interleaving the acoustic emission signal with the sample core temperature, the exact phase transformation temperatures can be assigned for the steel. To unambiguously assign the AE signals to specific processes in the material, further microscopy and metallographic analyses are required [17].

The main parameters monitored were the phase transformation times, i.e., the transformation of the steel crystal lattice from body centred cubic (BCC) to face centred cubic (FCC). At this point, it is most advantageous to carry out a further heat treatment from recrystallisations, such as various types of quenching. A further delay in the temperature results in significant changes in the microstructure and in the material’s mechanical properties. Figure 4, Figure 5 and Figure 6 show the graphical AE signal waveforms for the C45, C105W1, and 80CrV5 steels. Figures 9–11 on the left are the microstructures of the individual steels before AE analysis. This is predominantly a martensitic structure which is accompanied by ferrite and supra-deutectid carbides of various chemical compositions in the matrix.

The record of the acoustic emission event during the heating of C45 steel (Figure 4) shows two main areas. The first region is attributed to the martensitic structure decay, which shows the highest activity in the temperature range of 400 up to 500 °C (Figure 3). The decay of the martensitic structure is significantly affected by temperature. The primary inhibitor of AE in this case is dislocation motion. As time progresses, the annihilation of these processes occurs and hence the amplitude of AE decreases. Martensite characterized by its increased tensile strength and its presence in the microstructure significantly affects the mechanical properties of the steel. The decay of martensite is a process where carbon is precipitated into the ferrite to form iron carbide. The decay of the martensitic structure of this steel is shown by a well-defined region and is terminated after 47 s of heating. The heating of the sorbide structure formed by the martensite decay shows almost no AE events. The austenitic transformation response is measurable shortly before the austenitic temperature is reached and its length is directly proportional to the volume of the heated sample. In the austenitisation region, the main source of AE boundary migration events is the trans-formation from BCC to FCC. This transformation takes place between temperatures A1 and A3. The most prominent AE events are measurable at the start of the recrystallisation and are subsequently attenuated. In addition to carbon content, the AE event parameters are also significantly influenced by grain shape and size, grain boundary angles, dislocation density, and internal grain misorientation (which is related to the deformation of the crystal lattice within each grain).

Compared to the C45 steel (carbon content of 0.45%), the amplitudes of the acoustic emission events are significantly higher in the C105W1 steel samples, mainly from the martensitic decay region (see Figure 6). This is caused by the higher proportion of the martensitic structure, due to almost twice as much carbon content (1.0% C for the C105W1 steel). As already mentioned, in martensitic decay, the primary source of AE events is the dislocation motion. There is a tremendous amount of strain in martensite formation. As the carbon content increases, the chemical composition of martensite also changes proportionally. During the decay of martensite with a higher carbon content, more energy is released, and hence the amplitude of the AE signal increases. For this reason, there is also an earlier initiation of the martensite decay process. The presence of carbides in the steel microstructure also affects the length of the martensitic structure decay time. The microstructure of the quenched state consists of a matrix composed of martensite and finely dispersed globules of supra-eutectoid cementite. The size, distribution, and shape of the particles directly affect the local microstructure [18]. The authors [19] describe the carbide grain surroundings as a location with an increased risk of micro-cracking. The significant influence of carbides on phase transformations also lies in the formation of a barrier to dislocation motions. The subsequent phase transformation response, in this case, is clearly delineated and ends after 129 s of heating.

While the intensity of the martensitic structure decay is strongly influenced by the carbon content, the austenitisation length is almost identical for all of the analysed steels. For the 80CrV5 steel, the influence of other alloying elements, mainly chromium, on the acoustic emission response shape is evident. The authors [20] describe Cr, Mo, and V as some of the most important carbide-forming elements for carbon steels. Significantly, 80CrV5 steel only contains chromium at 0.65%. In terms of the mechanical properties, a microstructure containing primary carbides in its basic matrix appears to be the most advantageous [21]. As was found (based on metallographic analysis), the chemical composition fundamentally affects the intensity and shape of AE events, especially the carbon content. The 80CrV5 steel has a relatively high chromium content. Thus, part of the carbon is merged with chromium, where it is highly concentrated at the expense of the rest of the matrix in the form of chromium carbides of different chemical compositions [8,22]. Because of the highest chromium content, the highest density and size of chromium carbides is predicted for 80CrV5 steel.

According to the developed model, the following tables were possible to make, which was crucial for the further research in this field. The specific values of the temperatures (C45—A1: 700 °C; A3: 780 °C, C105W1—A1: 715 °C; ACM: 750 °C, 80CrV5—A1: 720 °C; ACM: 745 °C) and times of the observed phase transformations are given in Table 2. These are the average values of the tested samples that were monitored by the AE method:

### 3.3. Change in Hardness as a Function of the Austenitisation Time

Between the end of the martensitic structure decomposition and the beginning of the phase transformation, every used steel undergoes the following process. The structures decompose into supersaturated ferrite (with decreasing carbon content) and iron carbide.

Since significant changes in the microstructure and mechanical properties were expected as a function of the austenitisation time lag, a series of samples were created by loading them with different austenitisation time lags.

A heating time of 140 s was chosen as the reference value, when the austenitisation process is fully completed, according to the acoustic emission measurements. Additional heating times were chosen as 210, 300, 900, and 3600 s. The aim of the experiment is to verify the thermal degradation rate of carbon steels as a function of the heating time.

All of the samples were subsequently quenched. A water bath at a constant temperature of 20 °C was chosen as the cooling medium for all of the samples. A suitably chosen cooling environment is another important aspect that has a major influence on the resulting mechanical properties of the steel [23].

The hardness of the resulting samples was subsequently measured using the Rockwell hardness (HRC) method, according to EN ISO 6508-1. The measured values were then statistically processed and entered into Table 3.

The measured data show a decrease in the hardness of all of the tested steels, depending on the austenitisation time. The decrease averages to approximately 4 HRC/hr and, from a microstructural point of view, the changes are quite devastating and must necessarily affect any further component use. The authors [24] mention shot peening as one of the ways to improve the mechanical properties of the C45 steel surface. The surfaces of the samples for SEM analysis were ground, polished, and then etched with 3% nitric acid solution in alcohol. These changes were clearly visible in the 80CrV5 steel samples, where Figure 10 shows the microstructure of the quenched steel after 140 s of heating. The second figure (right) shows the same material after heating for 3600 s. A change in the distribution of the structural phases (mainly chromium carbides) is evident. According to the spectral analysis, the labelled (circles and arrows) carbides can be clearly identified as chromium carbides (see Figure 10). The other carbides are more likely to be iron carbides. Steels with a high chromium content are characterised by a significantly slower drop in hardness under thermal stress. It has been experimentally verified that 80CrV5 steel has the lowest hardness drop with increasing temperature compared to the other steels analysed. The effect of carbides was not analysed. In addition to chromium carbides, the structure also contains cementite carbides and other fine supra-deutectoid carbides that could not be accurately identified. The sample is also riddled with a large number of cracks on a microscopic level, which undoubtedly have an effect on the resistance to impact loading. These cracks are clearly caused by too long of a temperature delay and thus thermal degradation. For this reason, the effect of prolonged temperature delay on the resulting mechanical properties and microstructure of the tested steels was analysed. From the measured hardness values, it is evident that the length of the heating significantly affects the hardness of the samples. From the analysis carried out, it appears that, to achieve the highest hardness values and to achieve the desired microstructure (it is a fine-grained structure that makes the most out of the chemical potential of the steels in question, with as little residual austenite as possible. The presence of cracks of any origin is completely unacceptable. If the heating time is exceeded, the research carried out has shown the presence of micro-cracks), it is critical to terminate the heating immediately after the austenitisation process is completed. For all of the tested steels, there is an identical sharp initial drop in hardness after exceeding a heating time of 250 s; the HRC hardness curves are plotted in Figure 7.

In order to determine the statistical significance of the effect that the heating time has on the sample hardness of the evaluated steels, an analysis of variance (ANOVA) test [25,26] was performed using Tukey’s multiple comparison test (for a significance level of 95%). The graphical results of the ANOVA test are shown in Figure 8 and the results of Tukey’s test are shown in Table 4.

As the ANOVA test’s graphical results (Figure 8) and, in particular, the results of the multiple comparisons using Tukey’s test show, there was no statistically significant difference (at the 95% significance level) in the hardness for the C45 steel for the samples with heating times of 140 s and 210 s. However, there was a significant decrease in the hardness for the material with a heating time of 300 s.

The Figure 9, Figure 10 and Figure 11 represent samples quenched in the laboratory. The left part shows the microstructure with the experimentally determined heating time and the right part shows the microstructure with the heating time of 3600 s. The microstructure of C45 steel is shown in Figure 9. The different heating time manifested itself mainly in the morphology of the martensitic needles.

At the same time, after reaching a heating time of 300 s, the first occurrence of microcracks was observed for the 80CrV5 steel, and by 3600 s, the occurrence of microcracks was already observed over the whole analysed area of the sample (Figure 10, right). In terms of the subsequent accumulation of microstructural defects and crack propagation, the cyclic multi-axial loading of the component is the most dangerous [27].

Again, increasing the time to 900 and 3600 s does not lead to a significant decrease in the hardness. For the C105W1 and 80CrV5 steels, there is a statistically significant difference in the observed hardness for the specimens preheated at 140 s and 3600 s. For steel C105W1, a significantly lower carbide density was observed (heating time 3600 s). Although there were not quite significant changes like with steel 80CrV5, the presence of any cracks was not detected (see Figure 11). The changes in the hardness values are undoubtedly accompanied by changes in some other mechanical properties, such as the microhardness, tensile strength, and impact test.

## 4. Conclusions

The method and course of the heat treatment has a major influence on the final form of the steel. The difference between an ideally and incorrectly performed heat treatment is manifested both in a change in the mechanical properties and microstructure. Using non-destructive acoustic emission methods, the start and end time of phase transformations can be effectively monitored and accurately determined, and the resulting time applied to any subsequent heat treatment.

For the C45, C105W1, and 80CrV5 steel samples used in this experiment, the heating times required for the austenitic transformation were determined using AE methods. An optimum heating time between 130 s and 140 s was found for all of the samples, after which the AE signal no longer showed any significant events. Furthermore, the initial AE responses of the austenitic conversion were found to be 10–20 °C below the declared values given in the material standard. This deviation is most likely caused by variations in the chemical composition in the course of the steel production.

The hardness of all of the samples was measured identically, according to EN ISO 6508-1 using the HRC method. In terms of hardness, all of the samples show a rapid decrease in the hardness values during the first 250 s. The test results were subjected to an ANOVA and Tukey’s test for the statistical significance. For the C45 steel, there was a statistically significant difference (at a 95% significance level) in the decrease in the hardness of the samples with heating times exceeding 300 s. Additionally, for the C105W1 and 80CrV5 steels, there was a statistically significant difference between the measured hardness for the samples with heat loadings of 140 s and 3600 s. The overall decrease in the hardness ranges between 2.4 HRC for 80CrV5, 3.5 HRC for C105W1, and 3.7 HRC for C45 steel. Specifically, for the 80CrV5 steel, very significant changes that are incompatible with further use on machine parts were found in terms of the microstructure. The dependence of the exact austenitisation time on the mechanical properties was clearly demonstrated. As a consequence of the heating time, the phase structure modulation is associated with a reduction in the quality of the heat treatment and, thus, with the degradation of the mechanical properties. This statement can be clearly declared by the change in hardness of the tested steels.

## Figures and Tables

**Figure 1 materials-16-00518-f001:**
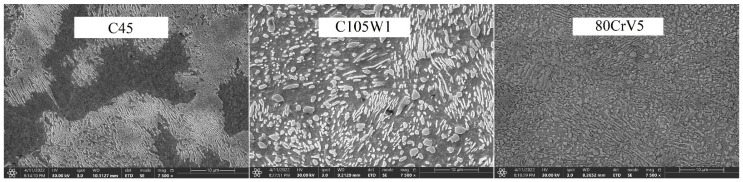
Natural states of the used steels.

**Figure 2 materials-16-00518-f002:**
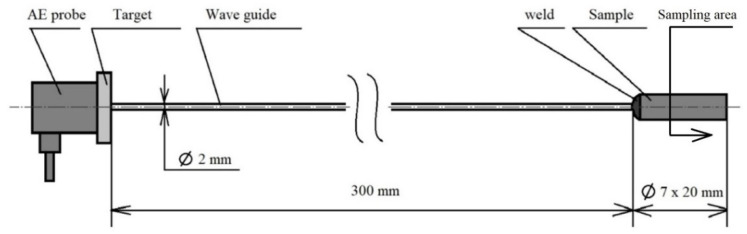
Test set-up.

**Figure 3 materials-16-00518-f003:**
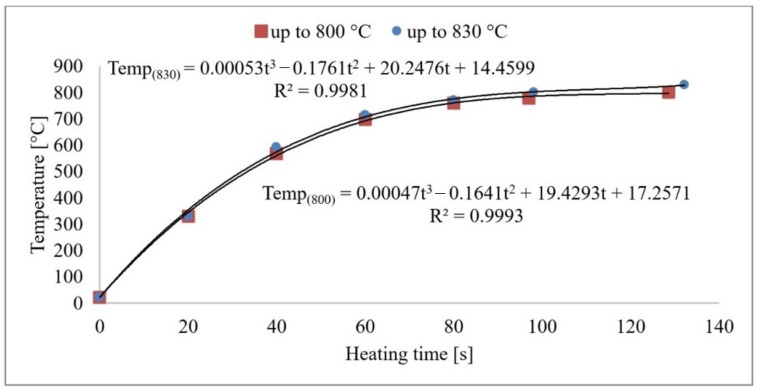
Sample core heating to 800 °C and 830 °C.

**Figure 4 materials-16-00518-f004:**
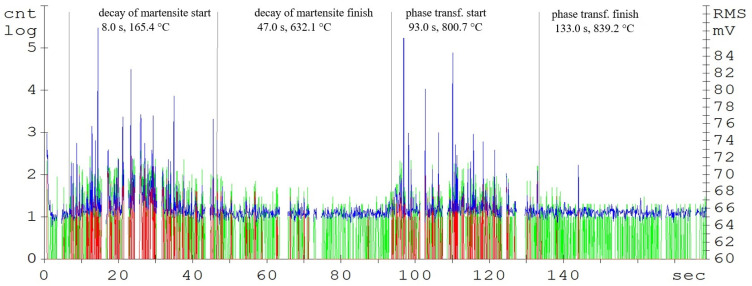
Acoustic emission spectrogram for the C45 steel samples, with a heating temperature of 830 °C. (Count 1: green, Count 2: red, RMS: blue).

**Figure 5 materials-16-00518-f005:**
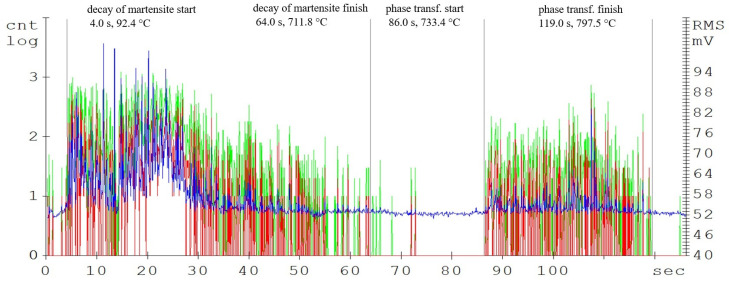
Acoustic emission spectrogram for the C105W1 steel samples, with a heating temperature of 800 °C. (Count 1: green, Count 2: red, RMS: blue).

**Figure 6 materials-16-00518-f006:**
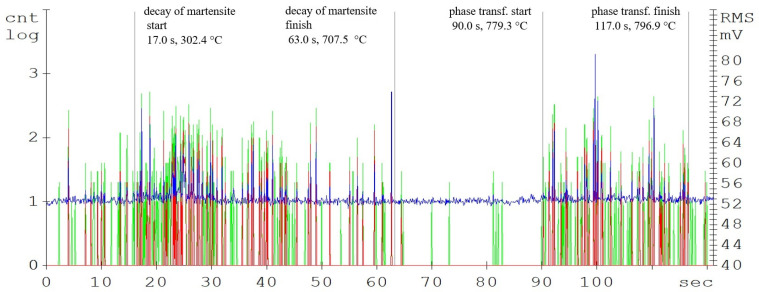
Acoustic emission spectrogram for the 80CrV5 steel samples, with a heating temperature of 800 °C. (Count 1: green, Count 2: red, RMS: blue).

**Figure 7 materials-16-00518-f007:**
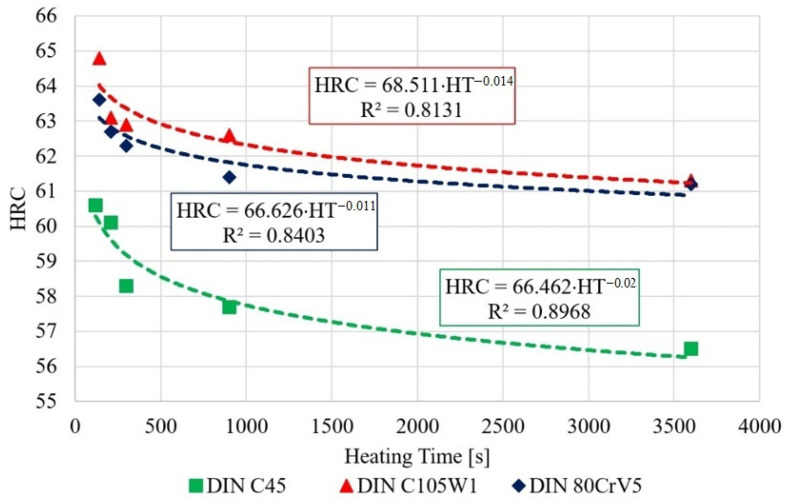
Decrease in the hardness depending on the heating time.

**Figure 8 materials-16-00518-f008:**
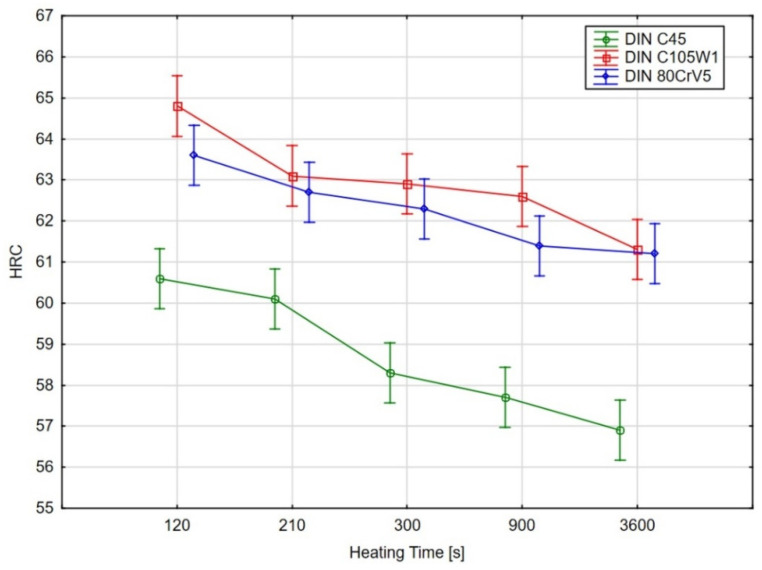
Graphical results of the ANOVA test for the significance of the effect of the heating time on the hardness, the points represent mean values, vertical bars show the 95% confidence intervals.

**Figure 9 materials-16-00518-f009:**
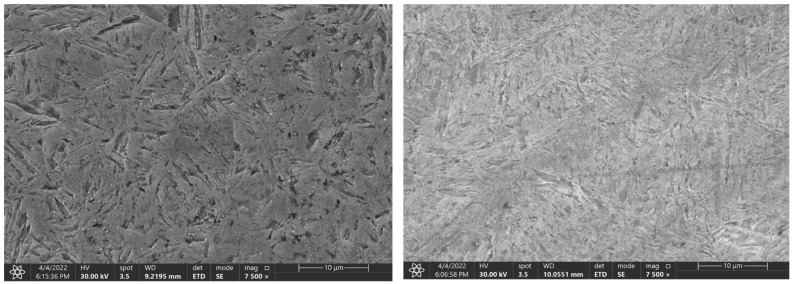
Change in the microstructure of the C45 steel, with a heating time of 140 s and 3600 s.

**Figure 10 materials-16-00518-f010:**
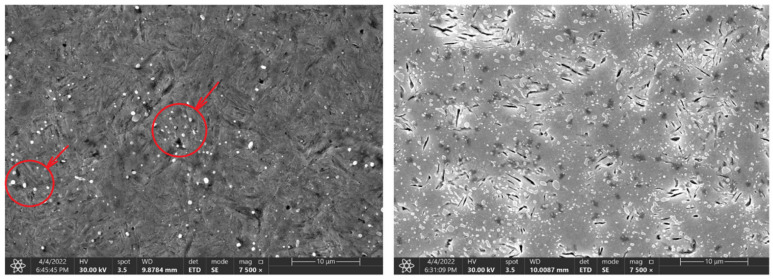
Change in the microstructure of the 80CrV5 steel, with a heating time of 140 s and 3600 s. Areas with chromium carbides are marked in the figure.

**Figure 11 materials-16-00518-f011:**
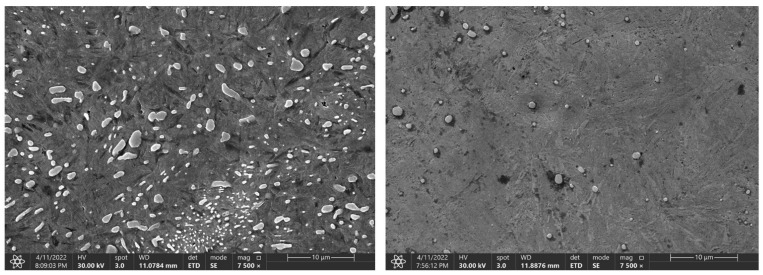
Change in the microstructure of the C105W1 steel, with a heating time of 140 s and 3600 s.

**Table 1 materials-16-00518-t001:** Chemical composition of the used steels.

C45					
C, wt. % Standard/Measured	Mn, wt. % Standard/Measured	Si, wt. % Standard/Measured	P, wt. % Standard/Measured	S, wt. % Standard/Measured	Cr, wt. % Standard/Measured
0.40–0.50/0.60	0.60–0.80/1.40	0.15–0.40/0.40	max. 0.35/-	max. 0.35/-	-
C105W1					
C, wt. % Standard/measured	Mn, wt. % Standard/measured	Si, wt. % Standard/measured	P, wt. % Standard/measured	S, wt. % Standard/measured	Cr, wt. % Standard/measured
0.95–1.09/0.90	0.20–0.35/0.40	0.15–0.30/0.50	max. 0.025/-	max. 0.30/-	max. 0.15/-
80CrV5					
C, wt. % Standard/measured	Mn, wt. % Standard/measured	Si, wt. % Standard/measured	P, wt. % Standard/measured	S, wt. % Standard/measured	Cr, wt. % Standard/measured
0.75–0.85/0.70	0.30–0.50/0.70	0.20–0.40/0.80	max. 0.030/-	max. 0.030/-	0.45–0.65/0.60

**Table 2 materials-16-00518-t002:** Phase transformations of the samples.

Steel	Decomposition of Martensite				Austenitisation			
	Time, s		Temperature, °C		Time, s		Temperature, °C	
	from	to	from	to	from	to	from	to
C45	8.0	47.0	165.4	632.1	93.0	133.0	800.7	839.2
C105W1	4.0	64.0	92.4	711.8	86.0	119.0	733.4	797.5
80CrV5	17.0	63.0	302.4	707.5	90.0	117.0	779.3	796.9

**Table 3 materials-16-00518-t003:** C45, C105W1, and 80CrV5 steel sample hardness at the selected heating times.

	C45			C105W1			80CrV5		
HT, s	Avg. MH, HRC	SD, HRC	VC, HRC	Avg. MH, HRC	SD, HRC	VC, HRC	Avg. MH, HRC	SD, HRC	VC, HRC
140	60.60	0.84	0.91	64.80	0.16	0.40	63.60	1.04	1.01
210	60.10	0.89	0.94	63.10	0.69	0.83	62.70	1.01	1.01
300	58.30	3.61	1.90	62.90	3.09	1.75	62.30	0.41	0.64
900	57.70	1.81	1.34	62.60	0.64	0.80	61.40	0.84	0.91
3600	56.90	1.29	1.13	61.30	0.81	0.90	61.20	1.36	1.16

HT—heating time; Avg. MH—average microhardness; SD—standard deviation; VC—coefficient of variation.

**Table 4 materials-16-00518-t004:** Tukey’s test results.

Heating Time, s	Steel	HRC	Groups						
3600	C45	56.9	a						
900	C45	57.7	a						
300	C45	58.3	a						
210	C45	60.1		b					
140	C45	60.6		b	c				
3600	80CrV5	61.2		b	c	d			
3600	C105W1	61.3		b	c	d			
900	80CrV5	61.4		b	c	d	e		
300	80CrV5	62.3			c	d	e	f	
900	C105W1	62.6				d	e	f	
210	80CrV5	62.7				d	e	f	
300	C105W1	62.9				d	e	f	
210	C105W1	63.1					e	f	g
140	80CrV5	63.6						f	g
140	C105W1	64.8							g

## Data Availability

Data is contained within the article.

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
