# Peer review of "Analysis of the Degradation of Pearlitic Steel Mechanical Properties Depending on the Stability of the Structural Phases"

_materials, 2023, doi:10.3390/ma16020518_

Round 1

Reviewer 1 Report

the idea to correlate microstructure variations to the AE signal parameters is interesting, some research was done in the past, not everything was successful because the microstructural variations are "quiet" for AE sensors, thus the experimental procedure should be carefully designed and presented in papers

the authors have collected valuable data that deserve publication, however the interpretation of "what was actually obtained here" can be improved

the tool steel microstructure is complex, it consists of martensite (a phase with high dislocation density), maybe pearlite (a phase consisting of Fe3C and ferrite), carbides of alloying elements (with different size distribution and number density depending on C and Cr content - taking the studied steels as an example).

Variations in the microstructural constituents (which happen during heating) may generate different values of AE parameters. In addition to improvements in Introduction, Methods, and Microstructure characterisation that might be recommended here, this manuscript does not clearly highlight two important points: (i) what microstructural parameters were considered as those that generate detectable AE signal and (2) what parameters of the AE signal were recorded in the experiments and why.

in general, this paper will be published, but it requires significant "data processing" and "text modification" prior to this

please see pdf file for more detailed comments

Author Response

Dear reviewer, We greatly appreciate your helpful comments and thank you for them. We tried to solve one by one and incorporate it into the manuscript. Details are given in the attached document (response to the reviewer's comments).

Reviewer 2 Report

This paper reports the effect of phase transformation and heating duration on the hardness of steel of three pearlitic steels. The acoustic emission technology is introduced to conduct nondestructive testing on the martensite phase transformation and recrystallization temperature range, and test the hardness of different steels with different heat preservation duration. The detection technology is novel and can provide a reference for related research. However, the overall quality of the manuscript was not high and there are some detailed aspects that need further modification.

1.Introduce

a. Important references in this field are missing. It is recommended to include some relevant literature and exposition on phase transition analysis and the effect of heat treatment heating time on the mechanical properties of steel.

2.Materials and Methods

a.  It is recommended that the steel composition content in Table 1 should be examined instead of the standard composition content.

b. In the part of materials and methods, there is only a test system. It is recommended to add information such as model of resistance furnace, hardness test and electron microscopethe and other relevant information..

c. It is recommended to include a graph of the sampling locations for hardness testing. The hardness test results can be affected if the sample is taken close to the weld.

3.Results and Discussion

In Table 3 and Table 4, there are some inconsistencies in font format, and there are some punctuation errors in the article, which need to be modified.

3.1a. If the heating rate is too fast in the experiment, which increases the thermal hysteresis, will the result of testing the core temperature of the sample cause a large error ?

3.2 a.  The recrystallization end time of C105W1 written in the paper is inconsistent with that marked in Figure 4, which needs to be modified.

3.3 a. Does the additional heating time mean to continue heating to raise the temperature, or to keep the temperature at this temperature after heating to 140 S ?

3.3 b.Why not show SEM pictures of C45 and C105W1 with different heating times ? Is there a phenomenon similar to microcrack? It is suggested to mark the content shown in Figure 8, such as microcracks, for the convenience of readers.

4.Conclusion

a. In the study, only the standard composition content of each steel was provided, but the composition of the steel used in the study was not analyzed. The conclusion that "This evolution is most likely caused by variations in the chemical composition in the course of the steel production." is not rigorous enough.

Author Response

(The authors gave the same response as above.)

Reviewer 3 Report

The acoustic emission method to determine the temperatures and times of phase transformations in tool steels was used in the work. The methodology is interesting. However, it is not clear what is the advantage of the acoustic emission method, for example, in comparison with dilatometric, DSC studies or electrical resistance measurements in this case? Why was such an experiment needed, if the conclusion about the best heat treatment modes is made on the basis of data on measuring the microhardness of steels after different heat treatment times. Other changes during heat treatment with different exposure times are not discussed, for example, dissolution/precipitation of the second phases particles, change in their shape and size, change in dislocation density, growth of austenite grains, etc.

Some comments:

1. Table 2. Is the steels elemental composition indicated in weight or atomic percent?

2. Line 63 "During the decay of this structure, the high content of the pearlitic structure in its natural state predetermines the good readability of the acoustic emission signal."

What structure are we talking about?

3. Why did you choose different heating temperatures for the studied steels?

4. "Figure 2. Sample core heating 800 °C and 830 °C."

May be «Sample core heating to 800°C and 830°C»?

5. Line134 "Based on a previous analysis, the following methodology was chosen: the aim of the research was to heat load the specimens with a pre-formed martensitic structure."

In what initial state were the steels, after what treatment? Please present the microstructure of steels in the initial state (before research).

6. Line 155 “The main parameters monitored were the recrystallisation times, i.e., the transformation of the steel crystal lattice from Body Centered Cubic (BCC) to Face Centered Cubic (FCC).”

What do you understand by the term recrystallization? How is it related to phase transformation?

7. What do you understand here by the decay of martensite? Please explain.

8. "Table 3. Sample hardness at the selected heating times."

For which steel are the results presented in Table 3?

9. Line 253 "A change in the distribution of the structural phases (mainly chromium carbides) is evident."

Please point with arrows where chromium carbides are shown in figure 8. Figure 8 shows a rather high density of particles of the second phases, although the chromium content in steel is only 0.45-0.65 %. What other carbides are present in steel? Cementite? How was the surface of samples prepared for SEM studies?

10. Line 253 “The sample is also riddled with a large number of cracks on a microscopic level…” Where did so many cracks come from? Please discuss this moment.

11. Line 264 "In order to determine the statistical significance of the effect that the heating time has 264 on the sample hardness of the evaluated steels, an analysis of variance (ANOVA) test was 265 performed using Tukey's multiple comparison test (for a significance level of 95%)."

Please add references to the literature that describe the methods for the ANOVA test and Tukey's multiple comparison test.

Author Response

(The authors gave the same response as above.)

Round 2

Reviewer 1 Report

the start and end of the martensite transformation to austenite could be correctly determined by the variation in AE signal, but the origin of the AE signal (i.e. what is happening in the microstructure during heating that causes AE signal) is discussed incorrectly

some comments in the pdf file may be useful for future work

Author Response

Response to the reviewer's comments is attached.

Reviewer 2 Report

This paper reports the effect of phase transformation and heating duration on the hardness of steel of three pearlitic steels. The acoustic emission technology is introduced to conduct nondestructive testing on the martensite phase transformation and recrystallization temperature range, and test the hardness of different steels with different heat preservation duration. The detection technology is novel and can provide a reference for related research. Meanwhile, some details of the manuscript need further revision.

1. There are some problems in text editing that need to be modified, such as two "was" in line 117.

2. The title of 3.1 has not been found in the article and needs to be supplemented.

3. It is recommended that the description of metallographic sampling locations in 3.3 be moved to Part II - Materials and Methods.

4. The picture of 80CrV5 steel should be Figure 9, and the line 398 in the paper is "Fig8, right", which needs to be corrected.

Author Response

(The authors gave the same response as above.)
